# Risk Factors for 30-Day Mortality in Nosocomial Enterococcal Bloodstream Infections

**DOI:** 10.3390/antibiotics13070601

**Published:** 2024-06-27

**Authors:** Verena Zerbato, Riccardo Pol, Gianfranco Sanson, Daniel Alexandru Suru, Eugenio Pin, Vanessa Tabolli, Jacopo Monticelli, Marina Busetti, Dan Alexandru Toc, Lory Saveria Crocè, Roberto Luzzati, Stefano Di Bella

**Affiliations:** 1Infectious Diseases Unit, Trieste University Hospital (ASUGI), 34125 Trieste, Italy; 2Clinical Department of Medical, Surgical and Health Sciences, Trieste University, 34129 Trieste, Italystefano.dibella@asugi.sanita.fvg.it (S.D.B.); 3Sports and Exercise Medicine Division, Department of Medicine, University of Padova, 35128 Padua, Italy; 4Medical Emergency Service, Trieste University Hospital (ASUGI), 34125 Trieste, Italy; 5Department of Health Science, Section of Anaesthesiology and Intensive Care, University of Florence, 50139 Florence, Italy; 6Microbiology Unit, Trieste University Hospital (ASUGI), 34125 Trieste, Italy; 7Department of Microbiology, “Iuliu Hațieganu” University of Medicine and Pharmacy, 400012 Cluj-Napoca, Romania; 8Liver Clinic, Trieste University Hospital (ASUGI), 34125 Trieste, Italy

**Keywords:** *Enterococcus faecium*, *Enterococcus faecalis*, bloodstream infection, bacteremia, mortality, vancomycin-resistant enterococci

## Abstract

Enterococci commonly cause nosocomial bloodstream infections (BSIs), and the global incidence of vancomycin-resistant enterococci (VRE) BSIs is rising. This study aimed to assess the risk factors for enterococcal BSIs and 30-day mortality, stratified by *Enterococcus* species, vancomycin resistance, and treatment appropriateness. We conducted a retrospective cohort study (2014–2021) including all hospitalized adult patients with at least one blood culture positive for *Enterococcus faecalis* or *Enterococcus faecium*. We included 584 patients with enterococcal BSI: 93 were attributed to vancomycin-resistant *E. faecium*. The overall 30-day mortality was 27.5%; higher in cases of BSI due to vancomycin-resistant *E. faecium* (36.6%) and vancomycin-sensitive *E. faecium* (31.8%) compared to *E. faecalis* BSIs (23.2%) (*p* = 0.016). This result was confirmed by multivariable Cox analysis. Independent predictors of increased mortality included the PITT score, complicated bacteremia, and age (HR = 1.269, *p* < 0.001; HR = 1.818, *p* < 0.001; HR = 1.022, *p* = 0.005, respectively). Conversely, male gender, consultation with infectious disease (ID) specialists, and appropriate treatment were associated with reduced mortality (HR = 0.666, *p* = 0.014; HR = 0.504, *p* < 0.001; HR = 0.682, *p* = 0.026, respectively). In conclusion, vancomycin-resistant *E. faecium* bacteremia is independently associated with a higher risk of 30-day mortality.

## 1. Introduction

The *Enterococcus* species are a common cause of nosocomial bacteremia. In the United States, enterococci are the first most common bacteria causing central line-associated bloodstream infections (BSIs) in long-term acute-care hospitals [1]. In 2019, the *Enterococcus* species ranked second among pathogens responsible for intensive care unit (ICU)-acquired BSIs in Europe [2]. 

Two species cause most of the enterococcal infections in humans: *Enterococcus faecalis* and *Enterococcus faecium* [3]. Other species involved in human infections are as follows: *E. casseliflavus*, *E. gallinarum*, *E. raffinosus*, *E. avium*, and *E. durans* [4,5]. Enterococci are not highly virulent bacteria. Typical enterococcal virulence factors are cytolysin, pili, gelatinase, aggregation substance, and extracellular surface proteins. These virulence factors contribute to the ability of enterococci to form biofilms [6]. Most of them are absent in *E. faecium*. Moreover, *E. faecalis* and *E. faecium* have different patterns of acquired antimicrobial resistance, and these are more frequently observed in *E. faecium*, such as with penicillins and glycopeptides [3]. Most *E. faecium* isolates exhibit ampicillin resistance, which is mainly due to the production of low-affinity penicillin-binding proteins (PBPs), especially PBP5. In contrast, high-level penicillin resistance in *E. faecalis* is much less common. While glycopeptide resistance can be found in both *E. faecalis* and *E. faecium*, it is more frequently associated with *E. faecium* [3,7,8].

Antimicrobial resistance (AMR) is known to be a leading cause of death around the world [9]. Recently, the European Antimicrobial Resistance Collaborators conducted a cross-country systematic analysis about bacterial antimicrobial resistance in the World Health Organization (WHO) European region in 2019. They estimated that 541,000 deaths were associated with bacterial AMR, and 47,200 of these were attributable to bloodstream infections. *E. faecium* was one of the seven pathogens responsible for most deaths associated with AMR [10]. AMR is also a serious threat to public health and national health systems. In this regard, nosocomial vancomycin-resistant enterococci (VRE) colonization [11] and infections significantly increase hospital costs [12].

A retrospective multicenter study, conducted in Italy from 2011 to 2017, found a progressive increase in the incidence of enterococcal bacteremia, and particularly those caused by vancomycin-resistant (VR) *E. faecium*. Resistance to ampicillin was detected in 6.8% and 89.1% of *E. faecalis* and *E. faecium* bacteremia cases, respectively. Resistance to vancomycin was detected in 1.3% and 14.1% of *E. faecalis* and *E. faecium* bacteremia cases, respectively. Resistance to tigecycline and linezolid was rarely observed [13]. 

Sources of enterococcal bacteremia usually are urinary and found in the gastrointestinal tract for community-acquired BSIs and intravascular and urinary catheters for hospital-acquired bacteremia [14]. In almost 20% of cases, the source of infection is not identified [15]. Most cases of enterococcal bacteremia are caused by *E. faecalis*, followed by *E. faecium* [15]. Polymicrobial bacteremia is often observed, ranging from 25% to 50% of enterococcal BSIs, depending on the study considered, and is usually associated with abdominal sources of infections [15,16]. The main risk factors for enterococcal BSIs are advanced age, immunosuppression, nosocomial infection under broad-spectrum antibiotics, prior enterococcal infections or colonization, recent surgery (mainly urinary or intra-abdominal), comorbidities related to urogenital and intra-abdominal organs, and presence of intravascular devices and/or indwelling urinary catheters [15].

Enterococcal BSIs are associated with high mortality rates, from 20% to 40% [15,17]. Before the approval of effective drugs for VRE strains, such as daptomycin and linezolid, two systematic reviews compared the outcomes of VRE versus vancomycin-sensitive *Enterococcus* (VSE) bacteremia. Both studies found an increased risk of mortality for VRE bacteremia (relative risk [RR], 2.38; 95% confidence interval [CI] 2.13–2.66; odds ratio [OR], 2.52; 95% CI, 1.87–3.39) [18,19]. In 2016, Prematunge et al. conducted a meta-analysis of 11 studies comparing VRE versus VSE bacteremia, confirming that VRE bacteremia is associated with an increased risk of in-hospital mortality and length of stay (LOS) [20]. Kramer et al., in 2018, in a retrospective cohort study on patients with enterococcal bacteremia, reported that in-hospital mortality and infection-attributed hospital stay are not influenced by vancomycin resistance but by the *Enterococcus* species (*E. faecium* is an independent risk factor for in-hospital mortality) [21]. More recently, a systematic review and meta-analysis found out a higher mortality for VR *E. faecium* bacteremia compared with vancomycin-sensitive (VS) *E. faecium* BSI (RR 1.46; 95% CI 1.17–1.82), while no difference was observed when comparing VR *E. faecium* vs. VR *E. faecalis* BSI (RR 1.00; 95% CI 0.52–1.93) [22]. Thus, according to the available studies, we cannot draw definitive conclusions about the outcome of enterococcal bacteremia.

The main aim of the present study is to investigate the risk factors for 30-day mortality for enterococcal BSIs, according to the *Enterococcus* species, resistance to vancomycin and appropriate treatment.

## 2. Results

During the study period, a total of 618 patients with enterococcal bacteremia were considered for inclusion. We excluded the following: 2 episodes for missing antimicrobial susceptibility test, 24 episodes because of species other than *E. faecalis* or *E. faecium* (10 *Enterococcus casseliflavus*, 5 *Enterococcus gallinarum*, 4 *Enterococcus avium*, 3 *Enterococcus durans*, 1 *Enterococcus hirae*, and 1 *Enterococcus raffinosus*), 5 episodes for concomitant *E. faecalis* and *E. faecium*, and 3 for missing species type.

Eventually, we included 584 patients with enterococcal bacteremia. Eleven patients had two separate episodes of enterococcal bacteremia (ten patients with the same species of *Enterococcus*, one patient had one *E. faecalis* BSI, and another one caused by *E. faecium*).

### 2.1. VRE Annual Prevalence

Only three *E. faecalis* BSIs, one observed in 2019 and two in 2020, were resistant to vancomycin. We excluded these patients from uni- and multivariate analysis due to the low number of strains isolated. A total of 93 vancomycin-resistant *E. faecium* were identified over the study period. The proportion of vancomycin-resistant *E. faecium* increased from 6.45% (n = 2/31) in 2014 to 51.06% (n = 24/47) in 2021 (Table 1). 

### 2.2. Population Analysis According to Species and Vancomycin Susceptibility

Demographic and clinical characteristics of the included patients according to *Enterococcus* species and resistance to vancomycin are described in Table 2.

The mean age of the total study population was 73.3 ± 12.2 years; 382 (65.2%) patients were males (65.2%).

We observed 340 episodes of vancomycin-sensitive *E. faecalis* BSI, 148 episodes of vancomycin-sensitive *E. faecium* BSI and 93 vancomycin-resistant *E. faecium* bacteremia. 

For the majority of cases of *E. faecalis* BSI, the source of infection was urologic, while an intra-abdominal focus was predominant in *E. faecium* BSIs (*p* < 0.001).

Hospital LOS before BSI diagnosis was significantly longer for vancomycin-resistant *E. faecium*, compared with other groups (22.3 ± 24.2 days vs. 13.3 ± 22.7 for VS *E. faecalis* and 16.6 ± 17.9 for VR *E. faecium*, *p* = 0.002). Previous chemotherapy and chronic immunosuppressive therapy were reported in 16.2% (*p* < 0.001) and 14.9% (*p* = 0.006) of VS *E. faecium* BSIs, respectively. Among patients receiving chronic immunosuppressive therapy (n = 59), seventeen were solid organ transplant recipients, and two were hematopoietic stem cell transplantation (HSCT) recipients. Only two patients had HIV infection. Between 2020 and 2021, nine patients with moderate/severe COVID-19 were diagnosed with enterococcal BSI. All of them received steroids during their hospital stay. Three patients received tocilizumab.

Infectious disease (ID) specialist consultation was conducted in 74.2% of vancomycin-resistant *E. faecium* bacteremia, compared with 41.5% and 42.6% of vancomycin-sensitive *E. faecalis* and *E. faecium* BSIs, respectively (*p* < 0.001).

No statistically significant difference was found in complicated BSI rates according to enterococcal species and vancomycin resistance (*p* = 0.055). Approximately 59.8% of females had complicated bacteremia compared to 53.9% of males. Infective endocarditis was largely caused by *E. faecalis* (n = 28, *p* = 0.014). 

Empirical therapy was initiated in 570 patients (98.1%). However, appropriate antimicrobial treatment commenced for only 350 patients (60.2%). Specifically, 65.9% of patients with vancomycin-sensitive *E. faecalis* bacteremia received appropriate antimicrobial therapy compared to 37.2% and 28% of those with vancomycin-sensitive *E. faecium* and vancomycin-resistant *E. faecium* BSIs, respectively (*p* < 0.001).

We observed only nine cases of *Clostridioides difficile* infection within 60 days of discontinuing therapy. Relapse of bacteremia within 60 days of discontinuing therapy occurred in 17 patients (for seven of them, source control was not performed).

We observed a total of 160 deaths within the 30 days following the first positive blood culture (BC) for *E. faecalis* or *E. faecium*. There were 79 deaths among patients with vancomycin-sensitive *E. faecalis* BSIs (30-day mortality rate: 23.2%), 47 in those with vancomycin-sensitive *E. faecium* BSIs (30-day mortality rate: 31.8%) and 34 in those with vancomycin-resistant *E. faecium* bacteremia (30-day mortality rate: 36.6%), *p* = 0.016.

### 2.3. Population Analysis According to Mortality

Demographic and clinical characteristics of patients and bacteria according to 30 day-mortality after BSI diagnosis are described in Table 3.

Regarding the ward upon BSI diagnosis, 23.6% of patients were hospitalized in the ICU. Only 54% of them survived, while 30-day mortality rates in medical and surgical wards were lower (24.7% and 16.3%, respectively; *p* < 0.001).

Complicated BSIs were observed in 65.4% of deceased patients compared to 52.4% of surviving patients (*p* = 0.004).

Appropriate antimicrobial treatment was started in 41.4% of deceased patients compared to 56.4% of surviving patients (*p* < 0.001). ID specialist consultation was performed in 273 patients (47% of the total). Approximately 79.9% of surviving patients received an ID consultation, while the survival rate decreased in the group that did not receive it (*p* < 0.001).

The 30-day mortality was analyzed through a multivariable Cox model (Table 4). Cox regression analysis confirmed an adjusted higher 30-day mortality rate for vancomycin-resistant *E. faecium* bacteremia compared to vancomycin-sensitive *E. faecium* BSIs and vancomycin-sensitive *E. faecalis* BSIs (Figure 1). The risk of death was higher for patients with complicated BSIs, higher PITT scores, and older age, while male gender, ID consultation, and appropriate antimicrobial treatment were predictive of lower mortality rates (Table 4).

For patients receiving an appropriate antimicrobial treatment, the sensitivity analysis confirmed a lowered risk for 30-day mortality only for patients with *E. faecalis* BSI (Figure 2).

## 3. Discussion

The burden of enterococcal BSIs is increasing worldwide [1]. Our country observed an increasing incidence of vancomycin-resistant *E. faecium* bacteremia over the last few years [13]. We calculated the annual prevalence of vancomycin-resistant strains on the total enterococcal BSIs reported at our Institution between 2014 and 2021. We observed 93 vancomycin-resistant *E. faecium* BSIs, and only 3 vancomycin-resistant *E. faecalis* BSIs. The prevalence of VR *E. faecium* bacteremia has increased from 6.45% in 2014 to 51.06% in 2021. However, most cases of BSIs were caused by *E. faecalis* (343 over a total of 584 bacteremia, 58.73%).

The origin of enterococcal bacteremia was identified mainly as urologic or intra-abdominal for *E. faecalis* and as intra-abdominal for *E. faecium* (both for VRE and VSE strains). The source of infection has not been identified in 19.4% of overall cases. These observations are in line with other cohort studies [23,24]. Hospital LOS before BSI diagnosis was significantly longer for patients with vancomycin-resistant *E. faecium* BSIs. This reflects the current evidence of a major involvement of *E. faecium* in nosocomial infections rather than the community-acquired ones [20,25].

Another interesting point concerns polymicrobial bacteremia. Usually 25–50% of enterococcal bacteremia are polymicrobial and have an abdominal origin [15,16]. In our study, we identified 187 cases of polymicrobial bacteremia, accounting for 31.9% of the total cases. In 99 instances, in addition to *Enterococcus*, bacteria from the *Enterobacteriaceae* family were also detected. The mortality rates for both polymicrobial and monomicrobial enterococcal bacteremia cases in our cohort were similar. As our data collection focused solely on therapies targeting *Enterococcus* species, we could not evaluate the appropriateness of antimicrobial treatment for the other isolated bacteria. However, our results are consistent with those reported by Lagnf et al. [16], in the only study known to us conducted with a focus on the outcome of polymicrobial enterococcal bacteremia.

Among major risk factors for enterococcal BSIs we have to consider advanced age, immunosuppression, and recent abdominal surgery [15]. Considering our cohort, older patients had a higher 30-day mortality rate (still significant in multivariable analysis). Chronic immunosuppressive therapy and previous chemotherapy were significantly associated with *E. faecium* bacteremia (both vancomycin-sensitive and resistant), as highlighted previously [26]. Recent abdominal surgery was not identified as a risk factor for enterococcal BSI and for 30-day mortality.

The risk factors for VRE bacteremia are mainly prior vancomycin use and VRE colonization [25]. Glycopeptide exposure before BSI diagnosis was not identified as a risk factor for enterococcal VR *E. faecium* BSI and for 30-day mortality in our cohort. At our institution, rectal swabs for the detection of VRE colonization are not routinely carried out in every ward. Thus, we did not analyze this variable. VR *Enterococcus faecium* colonization is also a risk factor for *C. difficile* infections in particular populations, such as HSCT recipients [27]. We reported only nine cases of CDI in our cohort. Consequently, we cannot stratify these data according to the *Enterococcus* species or vancomycin susceptibility.

Our study confirmed that the 30-day mortality rate was higher for vancomycin-resistant *E. faecium* BSIs (30-day mortality rate: 36.6%) compared with vancomycin-sensitive *E. faecium* BSIs (30-day mortality rate: 31.8%) and vancomycin-sensitive *E. faecalis* BSIs (30-day mortality rate: 23.2%). The multivariate analysis confirmed these observations. In particular, the risk of death was 1.5 and 2.0 times higher for vancomycin-sensitive and vancomycin-resistant *E. faecium* BSIs, respectively. The 30-day mortality rate in the group of patients who received an ID specialist dropped by 50% compared with the group who did not receive it (HR = 0.504, *p* < 0.001). These data are not new, but they reinforce the importance of bundles for the management of enterococcal BSI [28]. Male gender was associated with lower 30-day mortality (HR = 0.666, *p* = 0.014). Recently, a meta-analysis found a male/female ratio in VRE BSIs of 1.4 [29]. To the best of our knowledge, there are no other studies that have observed the protective role of being male in enterococcal BSIs. This result could be due to the higher proportion of complicated bacteremia in females compared to that in males. In the multivariate analysis, complicated bacteremia was associated with a higher 30-day mortality rate. There is not a consensus for the definition of enterococcal complicated bacteremia. We must mention two potential biases when considering our definition of complicated BSI. Firstly, given the retrospective design of our study, we were not able to properly assess positive follow-up blood cultures because they are not routinely performed by clinicians. Secondly, we considered all primary bacteremia and all bacteremia without control of infections’ sources as complicated.

While *E. faecium* is generally considered less virulent than *E. faecalis* [3], infections caused by *E. faecium* are associated with higher mortality rates and longer lengths of hospital stay [20], a trend corroborated by our study. Several potential explanations, though not conclusive, can be proposed. Firstly, the peculiar intrinsic and acquired antimicrobial resistance profile of *E. faecium* presents challenges in treatment [3]. Secondly, patients with *E. faecium* infections tend to be more medically fragile and have a higher burden of comorbidities [20]. Thirdly, given the relatively few therapeutic options for VRE BSIs, it is easier to miss the right empiric treatment compared to *E. faecalis* BSIs. Additionally, other less explored factors may contribute, such as disparities in biofilm formation between *E. faecalis* and *E. faecium* [30], as well as variations in host immune responses.

The multivariate analysis evidenced how an appropriate treatment was associated with a 30% reduction in the adjusted risk of death; however, this association was confirmed only for *E. faecalis* BSIs in the subgroup analysis. Inappropriate antibiotic therapy has already been identified as an independent risk factor for mortality in enterococcal bacteremia [31]. Recently, Russo et al. observed that starting an appropriate therapy for VRE bacteremia within 48 h from blood culture collection was independently associated with improved survival [32]. In this study, we did not assess the impact of different timing for appropriate treatment on 30-day mortality.

Our work has some limitations that deserve to be considered when interpreting the results. First, this is a retrospective study. Second, this study was conducted in only one hospital. Third, we did not consider active therapy against other isolated bacteria (for polymicrobial bacteremia) or stratify the analysis for different antimicrobial regimens. Finally, the definition we have chosen for appropriate therapy is arbitrary and not standardized. The strengths of this study are the long study period (from 2014 to 2021), the large sample (584 enrolled patients), and the fact that we made comparisons not only between VRE and VSE BSIs but also between *E. faecalis* and *E. faecium* BSIs.

Our future objectives are to enroll more patients through a multicentric study including enterococcal bacteremia from other Italian and European Hospitals, focus also on Enterococcus species other than *E. faecalis* or *E. faecium*, and assess the impact of different timing for appropriate treatment on 30-day mortality, according to different antimicrobial regimens.

## 4. Materials and Methods

### 4.1. Objectives of This Study

The aims of this study were as follows: (1) to calculate the annual prevalence of vancomycin-resistant strains on total enterococcal BSIs reported at our institution between 2014 and 2021; (2) to investigate risk factors for enterococcal BSIs, according to the *Enterococcus* species and resistance to vancomycin; and (3) to investigate risk factors for 30-day mortality, according to bacteria characteristics (*Enterococcus* species and resistance to vancomycin) and appropriate treatment.

### 4.2. Study Design and Population

We conducted a retrospective cohort study at Trieste University Hospital, Italy. All adult patients (aged > 18 years) hospitalized at our institution with at least one BC positive for *Enterococcus faecalis* or *Enterococcus faecium* during hospital stay were included. The study period ranges from 1 January 2014 to 31 December 2021.

Exclusion criteria were as follows: (1) pregnancy; (2) BCs positive for species other than *E. faecalis* and *E. faecium*; (3) lack of antimicrobial susceptibility test of isolated *Enterococcus*; (4) lack of *Enterococcus* typing; and (5) BCs positive for both *E. faecalis* and *E. faecium* at the same time. Additionally, duplicate BCs (up to 60 days following the last positive culture for the same *Enterococcus* spp.) from the same patient were excluded. 

### 4.3. Data Collection and Definitions

The following data were retrospectively collected from hospital electronic medical records: demographics (age and gender); comorbidities; chronic therapy; previous exposure to glycopeptides (vancomycin and teicoplanin); date of hospital admission; date of first positive BC; result of the in vitro susceptibility testing; source of infection; ICU admission and PITT score [33] at BSI diagnosis; ID specialist consultation; treatment prescribed for enterococcal BSI; evidence of polymicrobial bacteremia and complicated bacteremia; date of hospital discharge and date of death; relapse of bacteremia and *Clostridioides difficile* infection within 60 days of discontinuing therapy. We defined a new BSI caused by the same organism within 60 days of clinical and microbiological resolution of a previously treated BSI as a relapse. After 60 days, we considered the new BSI as a separate episode.

According to the Centers for Diseases Control and Prevention (CDC), possible sources of infection were classified as catheter-related, urologic, intra-abdominal, heart/cardiovascular devices, bone/skin/soft tissue, and unknown. In this last case, the bacteremia was defined as primary [34]. 

Enterococcal bacteremia was defined complicated when at least one of the following features were present: (a) infective endocarditis, (b) device-associated infection, (c) metastatic infection, (d) source control not done, (e) positive follow up BCs after 48–72 h, and (f) persistency of fever after 48–72 h from first positive BC. In the case of primary bacteremia, the source control was automatically defined as not documented and, therefore, not done. Enterococcal bacteremia was defined as polymicrobial when at least one non-enterococcal bacterial species was isolated from the same blood culture as *Enterococcus* spp. and met the CDC criteria for bloodstream infection [34]. 

Appropriate antibiotic therapy was defined as an active therapy against isolated bacteria started within 24 h from BSI diagnosis and continued for at least five days. Antibiotic therapy was defined as active in accordance with in vitro isolate susceptibility.

The treatment prescribed for enterococcal BSI was documented as follows: time to empiric therapy; time to pathogen-specific therapy; duration of pathogen-specific therapy.

Antibiotic regimens that we considered appropriate are listed in Appendix A.

All isolates were identified by MALDI-TOF mass spectrometry (bioMérieux, Marcy-l’Etoile, France), while antimicrobial susceptibility was assessed with the Vitek2 system (bioMérieux, Marcy-l’Etoile, France). Resistance to vancomycin was defined when a minimal inhibitory concentration > 4 was detected, according to the EUCAST criteria.

Mortality was defined as death of any cause within the 30 days following the first positive BC for *E. faecalis* or *E. faecium*. 

All data were pseudonymized via a web-based central, password-protected clinical database management system.

### 4.4. Statistics

Continuous variables were presented as means ± standard deviations (SD). The between-group comparisons were analyzed via Student’s *t* test for independent samples after determining whether or not equal variance could be attributed to the subgroups as per Levene’s test. Nominal variables were shown as a number and percentage, and the respective contingency tables were analyzed using χ test or Fisher’s exact test, as appropriate.

The prevalence of vancomycin resistance among *E. faecium* was calculated as the number of resistant strains over the total number of *E. faecium* isolates.

The 30-day mortality was analyzed according to bacteria characteristics (*Enterococcus* species and resistance to vancomycin) and appropriate treatment through multivariable Cox proportional hazards models with forward stepwise selection. The results were presented as an adjusted proportional hazard ratio (HR) and 95% confidence intervals (CIs). Aiming at examining the potential impact of survival bias among the patients receiving or not receiving an appropriate antimicrobial therapy, 30-day mortality was computed separately among bacteria subgroups for sensitivity analysis.

A *p*-value < 0.05 was set for statistical significance.

All statistical analyses were performed using the software IBM SPSS Statistics, version 24.0 (New York, NY, USA: IBM Corp.).

## 5. Conclusions

The mortality rate of enterococcal bacteremia is high. Our study confirms that vancomycin-resistant *E. faecium* bacteremia is independently associated with a higher risk of 30-day mortality, and delayed appropriate antimicrobial treatment is associated with a higher mortality rate. However, the employment of ID specialist consultation and appropriate antimicrobial therapy, along with patients’ male gender, were associated with significant lower mortality rates.

## Figures and Tables

**Figure 1 antibiotics-13-00601-f001:**
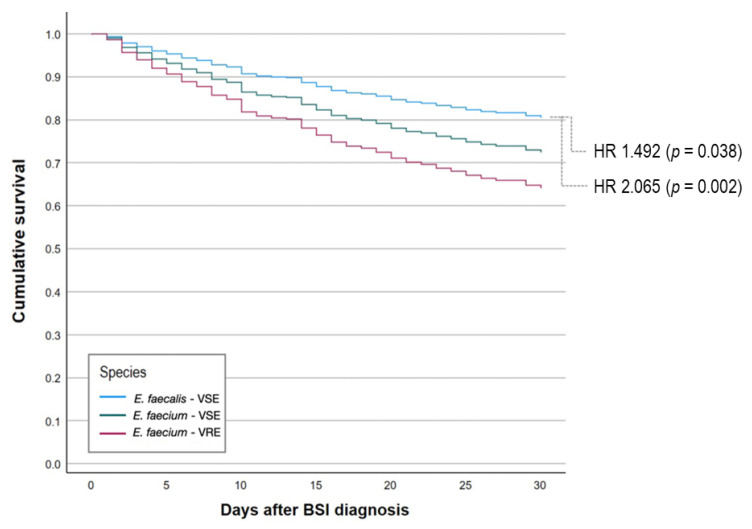
Adjusted Kaplan–Meier curves for the proportional risk of 30-day death in patients with vancomycin-sensitive *E. faecalis* BSIs, vancomycin-sensitive *E. faecium* BSIs, and vancomycin-resistant *E. faecium* BSIs. HR: hazard ratio. VSE: vancomycin-sensitive *Enterococcus*. VRE: vancomycin-resistant *Enterococcus*. BSI: bloodstream infection.

**Figure 2 antibiotics-13-00601-f002:**
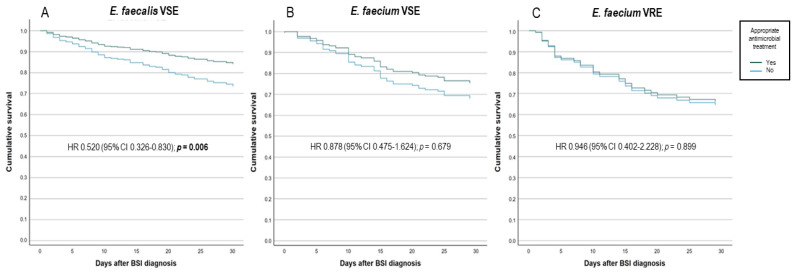
Adjusted Kaplan–Meier curves for the proportional risk of 30-day death in patients receiving or not receiving appropriate antimicrobial treatment according to the isolated bacteria (**A**) vancomycin-sensitive *E. faecalis*, (**B**) vancomycin-sensitive *E. faecium*, and (**C**) vancomycin-resistant *E. faecium*. HR: hazard ratio. CI: confidence interval. VSE: vancomycin-sensitive *Enterococcus*. VRE: vancomycin-resistant *Enterococcus*. BSI: bloodstream infection.

**Table 1 antibiotics-13-00601-t001:** Annual prevalence of vancomycin-resistant strains on total enterococcal BSIs.

Year	*E. faecium* BSIs	VR *E. faecium* BSIs	Annual Prevalence
2014	31	2	6.45%
2015	28	10	35.71%
2016	27	11	40.74%
2017	31	11	35.48%
2018	12	5	41.67%
2019	29	15	51.72%
2020	36	15	41.67%
2021	47	24	51.06%
All years (2014–2021)	241	93	

BSI: bloodstream infection. VR: vancomycin-resistant.

**Table 2 antibiotics-13-00601-t002:** Characteristics of patients according to bacterial species and vancomycin susceptibility.

Variable	*E. faecalis* VSE (n = 340)	*E. faecium* VSE (n = 148)	*E. faecium* VRE (n = 93)	*p*-Value
Age (years)	73.7 ± 12.2	73.8 ± 12.2	71.4 ± 12.2	0.222
Gender (male)	233 (68.5%)	91 (61.5%)	55 (59.1%)	0.131
Charlson Comorbidity Index	3.4 ± 2.5	3.4 ± 2.3	3.4 ± 2.4	0.998
Pitt bacteremia score	2.4 ± 2.3	2.0 ± 2.4	2.7 ± 2.1	0.074
Previous glycopeptides exposure	24 (7.1%)	4 (2.7%)	5 (5.4%)	0.159
Previous chemotherapy	17 (5.0%)	24 (16.2%)	15 (16.1%)	<0.001
Previous abdominal surgery	52 (15.3%)	35 (23.6%)	21 (22.6%)	0.052
Chronic immunosuppressive therapy	23 (6.8%)	22 (14.9%)	14 (15.1%)	0.006
Hospital LOS before BSI	13.3 ± 22.7	16.6 ± 17.9	22.3 ± 24.2	0.002
Ward at BSI diagnosis				0.056
Medical	177 (52.1%)	71 (48.0%)	43 (46.2%)	
Surgical	89 (26.2%)	46 (31.1%)	18 (19.4%)	
Intensive care unit	74 (21.8%)	31 (20.9%)	32 (34.4%)	
Source of infection				<0.001
Primary bacteremia	72 (21.2%)	27 (18.2%)	14 (15.1%)	
Bone/skin/soft tissue	17 (5.0%)	4 (2.7%)	5 (5.4%)	
Heart/cardiovascular devices	72 (21.2%)	41 (27.7%)	31 (33.3%)	
Intra-abdominal compartment	88 (25.9%)	63 (42.6%)	33 (35.5%)	
Urinary tract	91 (26.8%)	13 (8.8%)	10 (10.8%)	
Source control				0.027
No	77 (22.6%)	19 (12.8%)	14 (15.1%)	
Yes	173 (50.9%)	94 (63.5%)	59 (63.4%)	
Not documented	90 (26.5%)	35 (23.6%)	20 (21.5%)	
Polymicrobial BSI	118 (34.7%)	44 (29.7%)	24 (25.8%)	0.209
Infective endocarditis	28 (8.2%)	3 (2.0%)	3 (3.2%)	0.014
Complicated BSI	203 (59.7%)	71 (48.0%)	51 (54.8%)	0.055
ID specialist consultation	141 (41.5%)	63 (42.6%)	69 (74.2%)	<0.001
Interval BSI-empiric therapy (days) §	0.6 ± 1.3	0.6 ± 1.4	0.4 ± 1.1	0.554
Interval BSI-active therapy (days) ¥	1.0 ± 1.8	1.7 ± 2.0	2.4 ± 2.9	<0.001
Appropriate antimicrobial treatment	224 (65.9%)	55 (37.2%)	26 (28.0%)	<0.001
30-days mortality	79 (23.2%)	47 (31.8%)	34 (36.6%)	0.016

Data are reported as mean ± standard deviation or number (percentage). BSI: bloodstream infection. IDs: infectious diseases. LOS: length of hospital stay. VSE: vancomycin-sensitive *Enterococcus*. VRE: vancomycin-resistant *Enterococcus*. §: n = 570. ¥: n = 490.

**Table 3 antibiotics-13-00601-t003:** Demographic and clinical characteristics of patients and bacteria according to 30-day mortality.

Variable	Overall (n = 581)	Survived (n = 421)	Dead (n = 160)	*p*-Value
Age (years)	73.3 ± 12.2	72.5 ± 12.5	75.4 ± 11.1	0.006
Gender (male)	382 (65.2%)	288 (67.9%)	94 (58.0%)	0.024
Charlson Comorbidity Index	3.4 ± 2.4	3.3 ± 2.4	3.6 ± 2.4	0.188
Pitt bacteremia score	2.4 ± 2.3	2.0 ± 2.0	3.3 ± 2.7	<0.001
Previous glycopeptides exposure	34 (5.8%)	26 (6.1%)	8 (4.9%)	0.580
Previous chemotherapy	56 (9.6%)	35 (8.3%)	21 (13.0%)	0.083
Previous abdominal surgery	108 (18.4%)	84 (19.8%)	24 (14.8%)	0.163
Previous immunosuppressive therapy	59 (10.1%)	41 (9.7%)	18 (11.1%)	0.604
Species/vancomycin sensitivity				0.016
*E. faecalis*/VSE	340 (58.5%)	261 (76.8%)	79 (23.2%)	
*E. faecium*/VSE	148 (25.5%)	101 (68.2%)	47 (31.8%)	
*E. faecium*/VRE	93 (16.9%)	59 (63.4%)	34 (36.6%)	
Ward at BSI diagnosis				<0.001
Medical	291 (50.1%)	219 (75.3%)	72 (24.7%)	
Surgical	153 (26.3%)	128 (83.7%)	25 (16.3%)	
Intensive care unit	137 (23.6%)	74 (54.0%)	63 (46.0%)	
Source of infection				0.001
Primary bacteremia	113 (19.4%)	67 (59.3%)	46 (40.7%)	
Bone/skin/soft tissue	26 (4.5%)	20 (76.9%)	6 (23.1%)	
Heart/cardiovascular devices	144 (24.8%)	104 (72.2%)	40 (27.8%)	
Intra-abdominal	184 (31.7%)	133 (72.3%)	51 (27.7%)	
Urologic	114 (19.6%)	97 (85.1%)	17 (14.9%)	
Polymicrobial BSI	187 (31.9%)	133 (31.4%)	54 (33.3%)	0.648
Complicated BSI	328 (56.0%)	222 (52.4%)	106 (65.4%)	0.004
Interval BSI-empiric therapy (days) §	0.6 ± 1.3	0.6 ± 1.3	0.6 ± 1.4	0.980
Interval BSI-active therapy (days) ¥	1.4 ± 2.1	1.3 ± 2.1	1.4 ± 2.1	0.576
Appropriate antimicrobial treatment	306 (52.2%)	239 (56.4%)	67 (41.4%)	<0.001

Data are reported as mean ± standard deviation or number (percentage). BSI: bloodstream infection. VSE: vancomycin-sensitive *Enterococcus*. VRE: vancomycin-resistant *Enterococcus*. §: n = 570. ¥: n = 490.

**Table 4 antibiotics-13-00601-t004:** Results of Cox regression of 30-day mortality on study variables.

Variable	HR (95% CI)	*p*-Value
Gender (male)	0.666 (0.481–0.921)	0.014
Age (years)	1.022 (1.007–1.038)	0.005
Pitt Bacteremia Score	1.269 (1.192–1.350)	<0.001
Species/vancomycin sensitivity		
*E. faecalis*/VSE (reference)	1.000 (/)	
*E. faecium*/VSE	1.492 (1.022–2.180)	0.038
*E. faecium*/VRE	2.065 (1.307–3.264)	0.002
Complicated BSI	1.818 (1.304–2.535)	<0.001
ID specialist consultation	0.504 (0.352–0.719)	<0.001
Appropriate antimicrobial treatment Gender (male)	0.682 (0.488–0.955)	0.026
	0.666 (0.481–0.921)	0.014

HR: hazard ratio. CI: confidence interval. BSI: bloodstream infection. VSE: vancomycin-sensitive *Enterococcus*. VRE: vancomycin-resistant enterococcus. ID: Infectious diseases.

## Data Availability

The datasets analyzed during the current study are not publicly available but are available from the corresponding author upon reasonable request.

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
