# Peer review of "Risk Factors for 30-Day Mortality in Nosocomial Enterococcal Bloodstream Infections"

_antibiotics, 2024, doi:10.3390/antibiotics13070601_

Round 1
Reviewer 1 Report
Comments and Suggestions for Authors#well written and will provide significant insight to the scientific community
# i will please allow myself to mention some points that i think are helpful to tackle to improve the overall quality of data presentation and clarity:
1. in section 2.2, it is not clarified if duplicate cultures were counted or if only one positive culture per patient. moreover, if a patient had more than one episode of bacteremia, how were the 2 different episodes defined (for example, the duration considered between each positive blood culture in order to define it as a new episode adn not in fact a persistent bacteremia). it is mentioned in results separate episdoes, line 180, this needs to be clarified according to what mentioned above
2. the jump to vre is abrupt in line 182, there is no clear transition point throughout the results section, my advise would be to divide the results into sections such as general results/repartitions of species and origin of bacteremia and demographics for example and then to discuss repartition by either vancomycin susceptibility or by faecium vs faecalis, and then in a different subsection to mention mortality and other statistics
3.it should be mentioned earlier on that there were no or few vancomycin resistant e.faecalis cases because it confuses the reader to follow the flow
4. line 213, ID consult required. what is meant by required and what defined it? was it simply presence of ID consult or whas it related to complexity or severity of case and did this thus imply more risk of mortality, if so, this may be a bias or limitation in results
5.c.diff mentioned in line 225, this is an improtant and relevant result that is not mentioned as an outcome to be measured in the article, it is beneficial to shed some more light in this in the intro or in methods, it could be attractive for readers to know it will be included
6. in the conclusion, it is overstated that the incidence worlwide is increasing
however agree with all
i would love to review the second version of this article because the results are interesting but confusing to read and follow in the current format
thank you
Author Response
“In section 2.2, it is not clarified if duplicate cultures were counted or if only one positive culture per patient. moreover, if a patient had more than one episode of bacteremia, how were the 2 different episodes defined (for example, the duration considered between each positive blood culture in order to define it as a new episode adn not in fact a persistent bacteremia). it is mentioned in results separate episdoes, line 180, this needs to be clarified according to what mentioned above” - At the end of section 4.2, we added a clarification regarding duplicate cultures. At the beginning of section 4.3, we included additional definitions (relapse, second episode of BSI) for clarification.
“The jump to vre is abrupt in line 182, there is no clear transition point throughout the results section, my advise would be to divide the results into sections such as general results/repartitions of species and origin of bacteremia and demographics for example and then to discuss repartition by either vancomycin susceptibility or by faecium vs faecalis, and then in a different subsection to mention mortality and other statistics” - We updated the results as suggested.
“It should be mentioned earlier on that there were no or few vancomycin resistant e.faecalis cases because it confuses the reader to follow the flow” - We updated the results as suggested.
“line 213, ID consult required. what is meant by required and what defined it? was it simply presence of ID consult or whas it related to complexity or severity of case and did this thus imply more risk of mortality, if so, this may be a bias or limitation in results” - We changed the word “required with "performed". We simply mean that infectious disease consultation was requested (at our Institution, specialized consultations are at the discretion of the attending physician).
“c.diff mentioned in line 225, this is an improtant and relevant result that is not mentioned as an outcome to be measured in the article, it is beneficial to shed some more light in this in the intro or in methods, it could be attractive for readers to know it will be included” - In the methods section we mention this topic when we list the data collected (paragraph 4.3). We add a sentence also in the Discussion to give relevance to this topic.
“In the conclusion, it is overstated that the incidence worlwide is increasing” - We have removed the first two sentences of the paragraph, as suggested.
Reviewer 2 Report
Comments and Suggestions for Authors
I have read with interest the manuscript submitted by Zerbato et al, since AMR represent a global concern.
I have a few comments to be addressed in order to improve the quality of the manuscript:
- all abbreviations should be described at first use in the manuscript
- All Latin bacterial names should be italicized
- rows 48-50 the differences in the susceptibility pattern should be detailed
- the introduction should include a phrase containing the aim of the study
- the material and methods section should include some data on the microbiological methods used
- Results: table 1 should include the percentages as well as the total number of isolates identified each year. Also, the last row is not completed.
Any information about the treatment received? I suggest adding the antibiotic susceptibility rates as well.
Discussions section should be expanded and include further comparisons.
The reference list is scarce and not edited according to the mdpi pattern
Comments on the Quality of English Languagemainly typos/misspells of bacterial names
Author Response
“All abbreviations should be described at first use in the manuscript” - We have revised the text as requested.
“All Latin bacterial names should be italicized” - We have revised the text as requested.
“Rows 48-50 the differences in the susceptibility pattern should be detailed” - We detailed the susceptibility pattern and we added two new references.
“The introduction should include a phrase containing the aim of the study” - We have revised the existing sentence at the end of the paragraph by explicitly stating the main objective of the study.
“The material and methods section should include some data on the microbiological methods used” - we added data on the microbiological methods used at the end of the “Materials and Methods” paragraph.
“Results: table 1 should include the percentages as well as the total number of isolates identified each year. Also, the last row is not completed.” - Thank you for the comment. We completed the last row. The objective of the table is to show the annual prevalence of VRE strains (last column). Therefore, we prefer to keep the table as it is to make it easier to read.
“Any information about the treatment received? I suggest adding the antibiotic susceptibility rates as well.” - We have included the information regarding the treatment in the supplementary materials. Unfortunately, we are unable to provide additional data regarding susceptibility rates.
“Discussions section should be expanded and include further comparisons.” - We have expanded the discussion.
“The reference list is scarce and not edited according to the mdpi pattern” - We have arranged the bibliography according to the MDPI format and added some references.
Round 2
Reviewer 2 Report
Comments and Suggestions for Authors
I appreciate the author's efforts in addressing my comments. The quality of the manuscript has significantly improved and can be published after some minor corrections, such as:
Keep in mind that each abbreviation should be described separately in the abstract and main text (such as BSI)
The newly inserted references are not edited properly.
row 114 - it is worth completing the sentence with "due to the low number of strains isolated" and maybe mention just briefly if they were mainly from males/females/elderly or any other common thing that can be linked to the vancomycin resistance among E. faecalis.
Best regards,
Author Response
“Keep in mind that each abbreviation should be described separately in the abstract and main text (such as BSI)” - We have made the changes as requested.
“The newly inserted references are not edited properly.” - We have appropriately modified the new references.
“row 114 - it is worth completing the sentence with "due to the low number of strains isolated" and maybe mention just briefly if they were mainly from males/females/elderly or any other common thing that can be linked to the vancomycin resistance among E. faecalis.” - We have modified the sentence. The population (of 3) consisted of adult subjects with heterogeneous characteristics that we do not consider useful to report in the main text.